# Risk of Metachronous Neoplasia with High-Risk Adenoma and Synchronous Sessile Serrated Adenoma: A Systematic Review and Meta-Analysis

**DOI:** 10.3390/diagnostics13091569

**Published:** 2023-04-27

**Authors:** Umesha Boregowda, Chandraprakash Umapathy, Juan Echavarria, Shreyas Saligram

**Affiliations:** Division of Gastroenterology, Hepatology, and Nutrition, University of Texas Health San Antonio, San Antonio, TX 78229, USA

**Keywords:** colorectal cancer, serrated polyps, high-risk adenoma, metachronous neoplasm, high-grade dysplasia, surveillance colonoscopy

## Abstract

**Background:** Sessile serrated adenomas are important precursors to colorectal cancers and account for 30% of colorectal cancers. The United States Multi-Society Task Force recommends that patients with sessile serrated adenomas undergo surveillance similar to tubular adenomas. However, the risk of metachronous neoplasia when the high-risk adenoma co-exists with sessile serrated adenomas is poorly defined. **Objective:** To examine the risk of metachronous neoplasia in the presence of high-risk adenoma and synchronous sessile serrated adenomas compared with isolated high-risk adenoma. **Data sources:** PubMed, Embase, Scopus, Cochrane Library. **Study selection:** A literature search for studies evaluating the risk of metachronous neoplasia in patients with high-risk adenoma alone and those with synchronous high-risk adenoma and sessile serrated adenomas during surveillance colonoscopy was conducted on online databases. **Main outcome measures:** The primary outcome of interest was the presence of metachronous neoplasia. **Results:** Of the 1164 records reviewed, six (four retrospective and two prospective) studies met inclusion criteria with 2490 patients (1607 males, mean age 59.98 ± 3.23 years). Average follow-up was 47.5 ± 12.5 months. There were 2068 patients with high-risk adenoma on index colonoscopy and 422 patients with high-risk adenoma and synchronous sessile serrated adenomas. Pooled estimates showed a significantly elevated risk for metachronous neoplasia in patients with high-risk adenoma and synchronous sessile serrated adenomas (pooled odds ratio 2.21; 95% confidence intervals 1.65–2.96; *p* < 0.01). There was low heterogeneity (I^2^ = 11%) among the studies. Sensitivity analysis of the prospective studies alone also showed elevated risk of metachronous neoplasm (pooled odds ratio 2.56; 95%, confidence intervals 1.05–6.23; *p* = 0.04). **Limitations:** Inclusion of a small number of retrospective studies. **Conclusions:** The presence of high-risk adenomas and synchronous sessile serrated adenomas is associated with an increased risk of metachronous neoplasia. Therefore, shorter surveillance intervals may be considered in patients with high-risk adenoma and synchronous sessile serrated adenomas compared to those with high-risk adenoma alone.

## 1. Introduction

Colorectal cancer (CRC) is the second most common cause of mortality due to cancer and the third most common cancer worldwide, with an incidence of 1.8 million cases and 881,000 deaths annually, according to the 2018 global cancer statistics [1]. In the United States, it is the fourth most common cancer by incidence, and the third most common cause of cancer-related deaths [2]. The two types of precursor lesions for colon cancer are adenomatous polyps (conventional adenomas) and serrated lesions based on histology. Adenomatous colon polyps are the most common type of lesions and are responsible for approximately 70% of CRC [3,4]. Adenomatous polyps are found in 20–30% of screening colonoscopies for colon cancer [5,6,7]. All adenomatous polyps have dysplasia on histology, and if not removed, can evolve from low-risk adenomas to high-risk adenomatous polyps and, subsequently, into colon cancer [8]. The risk of CRC increases with high-risk adenomas [9]. Serrated lesions are classified into sessile serrated adenoma (SSA), hyperplastic polyps, and traditional serrated adenoma [10]. Compared to the rest of the serrated lesions, SSA leads to the majority of CRC and accounts for almost a third of colon cancers, making it a high risk for colon cancer [7,8]. SSPs are flat, pale lesions that may be covered with a mucus cap, which make it difficult to detect during colonoscopy [11]. They commonly occur in the proximal colon with prominent haustral folds, which makes them harder to detect. SSAs are found in 5–25% of the screening colonoscopies performed by expert endoscopists [12,13]. Individuals who had SSP with dysplasia had a 4.4% ten-year risk of CRC when compared to 2.3% in individuals with conventional adenoma alone [14].

Individually, we know the risk and rate of progression of adenomatous polyps and SSA to colon cancer [15,16]. It takes about 8–10 years for adenomatous polyps to transform into cancer [17]. This transformation can be quicker with high-risk adenomas [9]. Guidelines have been formulated, and appropriate surveillance intervals have been recommended based on this risk assessment [16,17]. However, we do not know the risk of CRC, especially in patients with high-risk adenoma and SSA, which contribute to the majority of CRC, with two different pathways of dysplasia development being present synchronously. This is especially important since the pathways that cause CRC by these two kinds of colon polyps are different [18]. The presence of BRAF mutations, CpG island methylation, and loss of immunoreactivity to MLH1 and MSH2 correlate with the clinical association of microsatellite-unstable CRC and SSPs. It is possible that the presence of these two different kinds of polyps can be aggressive in nature and accelerate malignant transformation, leading to interval cancer and, thereby, a higher risk of CRC.

The CRC risk can be estimated based on the risk of metachronous neoplasms detected during surveillance colonoscopy. The United States Multi-Society Task Force (USMSTF) recommends a surveillance colonoscopy every 3–7 years among individuals with SSPs or adenoma after index colonoscopy based on the number and size of the polyps [19,20]. The risk of metachronous neoplasia in individuals with high-risk adenoma and synchronous SSA is not clear. However, recent studies have shown an increased incidence of metachronous neoplasia in patients with high-risk adenoma and synchronous SSA.

There is currently no specific recommendation for the surveillance of colon polyps based on the synchronous presence of high-risk adenomas and SSA. These patients may need closer monitoring and earlier surveillance colonoscopies. Therefore, a systematic review and meta-analysis were performed to evaluate the risk of metachronous neoplasia in this population. In this systematic review and meta-analysis, we have compared the risk of metachronous neoplasm in individuals with high-risk adenoma, and synchronous SSA detected on index colonoscopy and those with high-risk adenoma alone. The aim of this meta-analysis is to provide knowledge regarding the risk for CRC in these high-risk populations, which will help in planning appropriate surveillance intervals to prevent CRC.

## 2. Materials and Methods

We adhered to the Preferred Reporting Items for Systematic Review and Meta-analysis (PRISMA) statement to perform this systematic review and meta-analysis [21].

### 2.1. Definitions

**High-risk adenoma:** This was defined as tubular adenoma ≥1 cm, or three or more adenomas, or an adenoma with villous histology or high-grade dysplasia.

**Low-risk adenoma:** This was defined as 1–2 tubular adenomas measuring <1 cm.

**Metachronous Neoplasia:** This was defined as recurrence of adenoma ≥1 cm (Tubular, Tubulovillous, Villous, High-Grade dysplasia), SSA recurrence, or CRC detected in surveillance colonoscopy.

### 2.2. Search Strategy

Electronic databases search was performed on PubMed, Embase, Scopus, and Cochran libraries for studies that compared the occurrence of metachronous neoplasms in patients who had high-risk adenoma alone or high-risk adenoma and synchronous SSA during screening colonoscopy. Two independent authors (CU and UB) reviewed the literature and collected the data on an electronic datasheet. A senior investigator (SS) independently reviewed the data whenever there were conflicts and verified the eligibility of the article. The following search words were used to find eligible articles on the databases: ‘adenoma’, ‘high-risk adenoma’, ‘tubular adenoma’, ‘tubulovillous adenoma’, ‘villous adenoma’, ‘sessile adenoma’, ‘sessile serrated adenoma’, ‘serrated adenoma’, ‘high-grade dysplasia’, ‘colonoscopy’, and ‘surveillance colonoscopy’. References of studies included in this meta-analysis were searched to find any relevant studies that were not detected in the database search.

### 2.3. Selection of Eligible Studies

Research articles that compared metachronous neoplasia during initial surveillance colonoscopy in patients who were previously diagnosed by biopsy-proven histology to have either high-risk adenoma and synchronous SSA or high-risk adenoma alone in index colonoscopy were included in the meta-analysis. Studies that were published as full-length articles only were included in this meta-analysis. Studies that were published as abstracts only, editorials, letters, reviews, case reports, and opinions were excluded. We also excluded studies with incomplete data, and when the histologic confirmation of colon lesions was not available. Studies that included individuals with increased risk of advanced neoplasm and CRC due to polyposis syndromes (Familial adenomatous polyposis, Peutz–Jeghers syndrome, Cowden’s syndrome, or Serrated polyposis syndrome) and inflammatory bowel disease were also excluded.

### 2.4. Data Collection

Screening of the articles and data collection were performed by two reviewers (CU and UB). Whenever there was uncertainty in study inclusion or exclusion of a study, a senior author (SS) reviewed the data independently, and a consensus decision was made. The collected data were tabulated on a Microsoft Excel sheet. The following information was collected about each included study: author, country of origin, the year of publication, type of study, the total number of study subjects, demographics, number of patients with high-risk adenoma alone, number of patients with high-risk adenoma, and synchronous SSA and number of patients with metachronous neoplasm in each group.

### 2.5. Outcomes

The primary outcome measure was the pooled odds ratio of metachronous neoplasm in the study groups. The biopsy result was considered the gold standard for the diagnosis of high-risk adenoma, SSA, and metachronous neoplasia. We also performed a subgroup analysis of retrospective and prospective studies to evaluate any difference in the risk of metachronous neoplasia in these groups. Furthermore, a sensitivity analysis was performed to examine the effect of individual studies on the pooled estimates by excluding one study at a time. Any significant difference in estimates was reported to reflect the same.

Secondary outcomes included pooled odds ratio for the risk of metachronous neoplasia in high-risk adenoma alone versus low-risk adenoma alone, high-risk adenoma with synchronous SSA versus low-risk adenoma with synchronous SSA, and high-risk adenoma alone versus low-risk adenoma with synchronous SSA.

### 2.6. Quality Assessment

The quality of each study included in this meta-analysis was evaluated utilizing the Risk of Bias Assessment tool for nonrandomized studies (RoBANS). Each study was scored for selection bias, measurement of exposure, blinding of outcome assessment, incomplete outcome data, and selective reporting.

### 2.7. Statistical Analysis

The pooled odds ratio of metachronous neoplasia, with a 95% confidence interval (CI), was calculated using the random-effects model. Heterogeneity among the included studies was assessed using the inconsistency index (I^2^). Heterogeneity of 25%, 50%, and 75% was considered low, moderate, and substantial heterogeneity, respectively. Categorical variables were reported in proportions, and continuous variables were reported as mean with standard deviation (SD). Revman 5.3 software (Review Manager Version 5.3; The Nordic Cochrane Centre, Copenhagen, Demark, The Cochrane Collaboration 2015) was used to perform statistical analysis.

## 3. Results

### 3.1. Study Selection

A total of 1160 articles were found for the search words after removing the duplicates. We found six eligible studies based on the inclusion and exclusion criteria for the primary outcome of interest [22,23,24,25,26,27]. Two studies were excluded from meta-analysis due to incomplete information [14,28]. All six studies reported data on low-risk adenoma alone, high-risk adenoma alone, and high-risk adenoma with synchronous SSA for pooled estimation of metachronous neoplasm. Only four studies reported data on metachronous neoplasm in individuals who had both low-risk adenoma with SSA and high-risk adenoma with synchronous SSA [23,24,25,26].

### 3.2. Study Characteristics

Among the six studies included in the meta-analysis, there were two multicenter studies and four single-center studies. There were two prospective studies and four retrospective studies. The mean duration of the study was 59.66 months (median 35.5; interquartile range: 32–137). Figure 1 shows the search strategy and selection process of eligible studies.

Table 1 lists all the studies that are included in the meta-analysis. All patients included underwent a screening colonoscopy for CRC and a surveillance colonoscopy based on the presence of high-risk adenoma and SSA with a mean interval of 47.5 ± 12.5 months between the two procedures. Quality assessment of included studies using RoBANS is shown in Appendix A. Publication bias was not performed since the number of studies included was less than 10.

### 3.3. Patient Characteristics

Among the six included studies, there were 5628 patients who underwent initial screening colonoscopies. Among them, a total of 2490 individuals, including 1607 (65%) males and 883 (35%) females, who underwent screening and subsequent surveillance colonoscopies were included in the pooled estimates of primary outcomes, had either high risk adenoma, or SSAs, or both. The mean age of the population was 59.98 ± 3.23 years. There were 2068 individuals with high-risk adenoma found on index colonoscopy. Among them, 422 patients had high-risk adenoma and synchronous SSA. Five of the six studies included individuals with a family history of CRC (*n* = 267), and all six studies (*n* = 662) reported a history of current or prior tobacco use.

### 3.4. Primary Outcome

The pooled odds ratio of metachronous neoplasia in patients with high-risk adenoma and synchronous SSA, when compared to high-risk adenoma alone, was significantly higher; pooled OR 2.12 [95% CI 1.63–2.76; *p* < 0.01]. Figure 2 shows the forest plot analysis comparing the pooled odds ratio for risk of metachronous neoplasm in individuals with high-risk adenoma and synchronous SSA and those who had high-risk adenoma alone. There was a moderate heterogeneity among the studies included in the meta-analysis [I^2^ = 11%].

Sensitivity analysis by the exclusion of one study at a time showed no significant change in the pooled estimates. A subgroup analysis of prospective and retrospective studies also showed that there was an increased risk of metachronous neoplasm in individuals with high-risk adenoma and synchronous SSA with a pooled OR of 2.56 [95% CI 1.05–6.23], *p* = 0.04, and pooled OR of 2.27 [ 95% CI 1.56–3.31], respectively. Figure 3 shows a forest plot for the pooled estimates of prospective and retrospective studies.

### 3.5. Secondary Outcomes

The pooled odds ratio also suggested an increased risk of metachronous neoplasm in individuals with high-risk adenoma and synchronous SSA when compared to low-risk adenoma with synchronous SSA [*n* = 902, pooled OR 3.54; 95% CI 1.83–6.86, *p* < 0.01]. Figure 4 shows a forest plot for pooled estimates of metachronous neoplasia in those with high-risk adenoma with synchronous SSA and those with low-risk adenoma with synchronous SSA.

The pooled estimates also suggested the risk of metachronous neoplasm is higher in individuals with high-risk adenoma alone when compared to individuals with low-risk adenoma only [*n* = 4844; pooled OR 3.70; 95% CI 1.82–7.51; *p* < 0.01]. Figure 5 shows a forest plot analysis for the high-risk adenoma alone versus low-risk adenoma alone. However, there was no significant difference in the risk of metachronous neoplasia between individuals with high-risk adenoma alone and those with low-risk adenoma with synchronous SSA (*n* = 1544; pooled odds ratio 2.18 (0.75–6.37); *p* = 0.15). Appendix A shows the forest plot analysis for pooled estimates for high-risk adenoma alone and low-risk adenoma with synchronous SSA.

## 4. Discussion

Screening colonoscopies for CRC prevention have been performed since the 1970s. However, since the 1980s, the rate of people being diagnosed with CRC has dropped. This was possible mainly because a greater number of people are screened with routine screening colonoscopies. From 2011 through to 2021, the incidence of CRC has decreased by 1% each year [2]. Further demographic studies suggested that the drop was mostly seen in the older population. However, the incidence of CRC has increased by 1–2% per year since the mid-1990s in younger adults who are <50 years old. Therefore, the USMSTF recently reduced the age for CRC screening to 45 years. The risk of CRC in any individual depends not only on age but also on other risk factors. Age and family history of CRC are non-modifiable risk factors, whereas smoking, alcohol use, overweight or obesity are modifiable risk factors. Therefore, appropriate screening strategies based on risk factors and findings from prior colonoscopies are the need of the hour. Previous studies have suggested higher risk of CRC in patients with advanced adenomas. Advanced adenoma is a distinct type of colorectal polyp defined by its size (adenomas measuring ≥ 10 mm) and histology (adenomas with villous histology or high-grade dysplasia). These patients have a higher risk of developing CRC. First-degree relatives of patients with advanced adenomas also have a significantly higher risk of CRC and, therefore, patients with advanced adenomas and their first-degree relatives warrant early CRC screening compared to the general population. Therefore, society guidelines suggest starting CRC screening 10 years earlier than the general population with an average risk.

This meta-analysis showed that the presence of high-risk adenoma and synchronous SSA on index colonoscopy is associated with a two-fold increased risk in metachronous neoplasia when compared to those with high-risk adenoma alone [*n* = 4653; pooled OR 2.12 [95% CI 1.63–2.76; *p* < 0.01]. Secondary analysis revealed an increased risk of metachronous neoplasia in individuals who had high-risk adenoma with synchronous SSA when compared to low-risk adenoma with synchronous SSA [*n* = 902, pooled OR 3.54; 95% CI 1.83–6.86, *p* < 0.01]. The risk of metachronous neoplasia in those with high-risk adenoma alone was higher when compared to individuals with low-risk adenoma alone [*n* = 4844; pooled OR 3.70; 95% CI 1.82–7.51; *p* < 0.01]. However, there was no significant difference in the risk of metachronous neoplasm between those with high-risk adenoma alone when compared to individuals with low-risk adenoma with synchronous SSA (*n* = 1544; pooled odds ratio 2.18 (0.75–6.37); *p* = 0.15).

The chromosomal instability pathway leads to CRC from high-risk adenoma, and the serrated pathway leads to CRC from SSA [29]. Therefore, the risk of CRC in individuals with SSA and high-risk adenoma is well known [20]. However, the risk quantification and its implications in patients with high-risk adenoma and synchronous SSA are currently not known. Many individuals are found to have a concurrent presence of polyps from both the adenomatous and the serrated pathway during colonoscopy. Recent literature indicates that individuals with high-risk adenoma and synchronous SSA have a higher risk of metachronous neoplasia [22,24]. In a recently published retrospective study on screening colonoscopies performed between 2000 and 2014, the presence of high-risk adenoma and synchronous SSA was shown to be an independent predictor of metachronous neoplasia [hazard ratio 3.20; 95%CI 1.31–7.82; *p* < 0.01]. The etiology of this increased risk is unclear, though it has been postulated that the presence of more than one oncogene pathway may have a role in it [10]. However, a meta-analysis evaluating the pooled estimates for the risk of CRC in this patient population is lacking.

Current USMSTF recommendations for CRC screening recommend a surveillance colonoscopy in 7–10 years from index colonoscopy for low-risk adenoma and 3–5 years for high-risk adenoma based on the size and number of polyps [19]. However, these recommendations do not account for the presence of high-risk adenoma and synchronous SSA. The data are limited to making recommendations for a surveillance strategy for CRC prevention in this group of patients. Our data support the need for close monitoring of these individuals with high-risk adenoma and synchronous SSA with shorter surveillance intervals.

Most recently, there was a similar meta-analysis that evaluated the risk of metachronous serrated lesions in patients with SSA on baseline colonoscopy and had more studies included in it [30]. However, we have focused on high-risk group, evaluating metachronous lesions in high-risk adenoma and synchronous SSA. These results are pertinent and can help identify individuals at increased risk for developing metachronous neoplasia in the future and provide further evidence for gastroenterologists and policymakers to come up with surveillance strategies tailored to prevent progression to CRC in this high-risk group.

Our results also support the fact that the risk of metachronous neoplasia is higher when high risk adenomas with synchronous SSA are present. However, there was no change in the risk of metachronous neoplasia when low risk adenomas and synchronous SSA were present, suggesting the risk of metachronous neoplasia is particularly dependent on high-grade dysplasia found in adenomatous polyps. When there is a synchronous presence of high-grade dysplasia in an adenoma and a second pathway with SSA, the risk of metachronous neoplasms is significantly higher.

Our meta-analysis has some inherent limitations; we only have a small number of studies with only 4653 patients. Most studies included in this meta-analysis are retrospective studies. Retrospective studies can significantly increase the risk of selection bias, thereby affecting the reliability of the results. The risk of metachronous neoplasms cannot be reliably estimated using observational studies alone. The quality of bowel preparation during initial screening and the adenoma detection rate of endoscopists who perform the screening colonoscopies can greatly affect the rate of metachronous neoplasia. Therefore, the increased risk of advanced neoplasia in the study population can be attributed to missed lesions and poor bowel preparation. It should also be noted that the included studies did not provide granular details on the adenoma detection rate of endocospists who participated in the studies or the quality of bowel preparation on index colonoscopy of surveillance colonoscopy. In this meta-analysis, we were unable to evaluate the adequate time interval between the index colonoscopy and surveillance colonoscopy for the risk stratification of these individuals. The risk of metachronous neoplasia can also be affected by other risk factors for CRC, including alcohol, tobacco use, obesity, family history of advanced adenomas, and other confounding factors. Our study could not assess the role of these risk factors on the risk of metachronous neoplasia in patients included in the studies. Moreover, an accurate measure of cancer risk is to calculate the pooled risk of CRC. The included studies did not report the occurrence of CRC in the study population and, therefore, we could not calculate the same. In addition, the total number of polyps in the included population can play a role in the development of subsequent CRC. However, this data was not reported in the included studies. Although, our study has several limitations, the results clearly suggest there is increased risk for development of metachronous neoplasia in individuals who have synchronous presence of high-risk adenomas and SSAs, which can indirectly suggest an increased risk for CRC.

Therefore, future studies should consider evaluating the long-term risk of metachronous neoplasia in individuals with high-risk adenoma and synchronous SSA on screening colonoscopy, and the possibility of an increased risk of progressing to CRC in prospective studies with a larger study population. Future studies should also attempt to evaluate the risk of metachronous neoplasia in the first-degree relatives of patients who have advanced adenomas alone and those with synchronous SSAs as well. Most importantly, the studies should evaluate the interobserver difference in detection of advanced adenomas based on endoscopists adenomas detection rate, quality of bowel preparation, and other risk factors for CRC such as smoking, alcohol, family history of CRC, and obesity.

## 5. Conclusions

In conclusion, the risk of metachronous neoplasm in individuals with high-risk adenoma and synchronous SSA is significantly higher when compared to those with high-risk adenoma alone. These individuals may be at higher risk for CRC. Close monitoring with short-interval surveillance colonoscopy plays a crucial role in the early detection and treatment of metachronous neoplasia, thereby preventing CRC among these individuals. Larger, longitudinal follow-up studies of individuals who have high-risk adenoma and synchronous SSA on index colonoscopy are needed to quantify the risk of CRC in this population and establish appropriate surveillance strategies. If possible, these studies should also assess the role of modifiable risk factors, such as alcohol use, tobacco use, and family history of metachronous neoplasia and CRC.

## Figures and Tables

**Figure 1 diagnostics-13-01569-f001:**
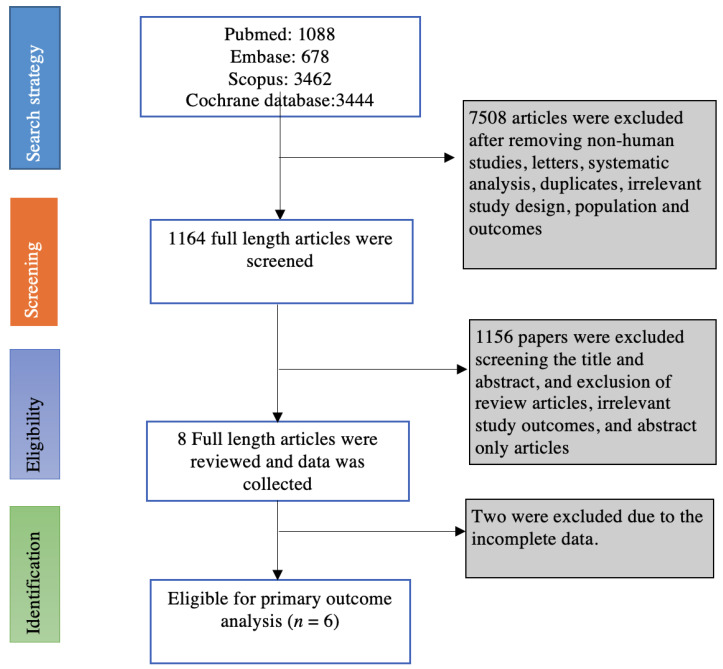
Flow chart of selecting eligible studies for the meta-analysis.

**Figure 2 diagnostics-13-01569-f002:**
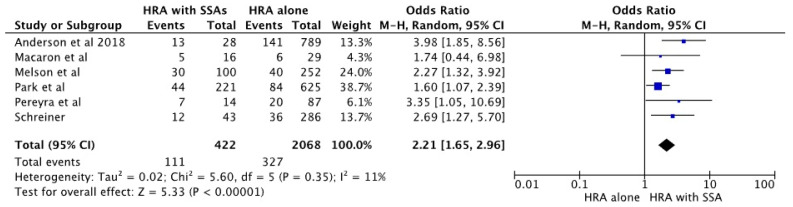
Forest plot of studies comparing pooled estimation of metachronous neoplasia in high-risk adenoma (HRA) with synchronous SSA versus high-risk adenoma alone [22,23,24,25,26,27].

**Figure 3 diagnostics-13-01569-f003:**
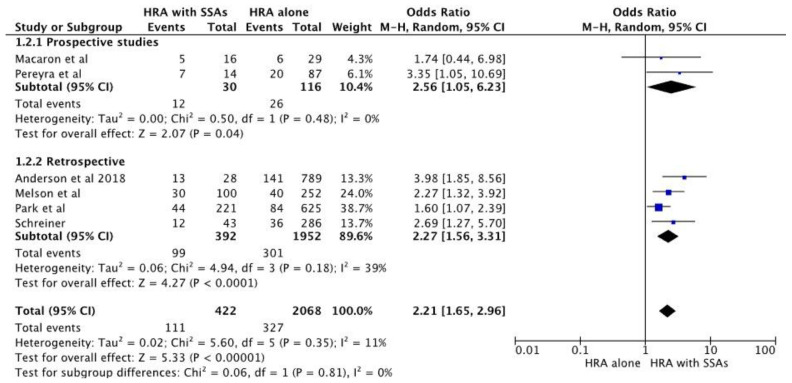
Forest plot of subgroup analysis of studies comparing pooled estimation of metachronous neoplasia in prospective and retrospective studies [22,23,24,25,26,27].

**Figure 4 diagnostics-13-01569-f004:**
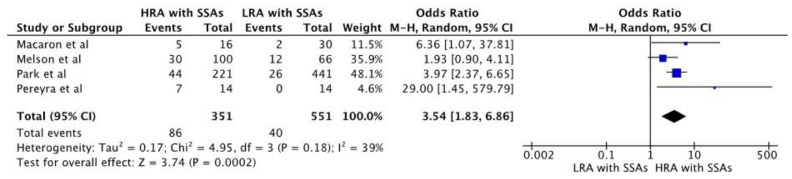
Forest plot of studies comparing pooled estimation of metachronous neoplasia in high-risk adenoma (HRA) with synchronous SSA vs. Low-risk adenoma (LRA) with synchronous SSA [23,24,25,26].

**Figure 5 diagnostics-13-01569-f005:**
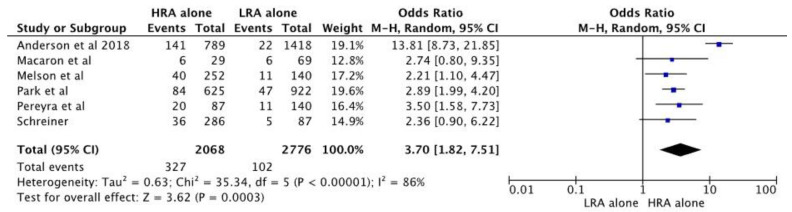
Forest plot of studies comparing pooled estimation of metachronous neoplasia in individuals with high-risk adenoma (HRA) alone and low-risk adenoma (LRA) alone [22,23,24,25,26,27].

**Table 1 diagnostics-13-01569-t001:** Characteristics of studies included in the meta-analysis.

Author	Study Design	Study Duration (In Months)	Number of Patients	Mean Age (Year)	Males (*n*)	Mean Surveillance Interval (Months)	High-Risk Adenoma with SSA on Index Colonoscopy (*n*)	HRA without SSA on Index Colonoscopy (*n*)	AN on Surveillance Colonoscopy in HRA + SSA	AN in HRA Alone on Surveillance
Anderson et al. [22]	Retrospective	137	817	61	406 (50%)	58.8	28 (3.5%)	789 (96.5%)	13 (46%)	141 (18%)
Macaron et al. [23]	Prospective	36	45	56.6	29 (64%)	30.9	16 (36%)	29 (64%)	3 (19%)	6 (21%)
Melson et al. [24]	Retrospective	83	352	61	191 (54%)	41.3	100 (28%)	252 (72%)	30 (30%)	40 (16%)
Park et al. [25]	Retrospective	35	846	56.3	609 (72%)	36	221 (26%)	625 (74%)	44 (20%)	84 (13%)
Pereyra et al. [26]	Prospective	32	101	65	51 (50%)	52	14 (14%)	87 (86%)	7 (50%)	20 (23%)
Schreiner et al. [27]	Retrospective	35	329	NA	321 (91%)	66	43 (13%)	286 (87%)	11 (26%)	36 (13%)

HRA: High-risk adenoma, SSA: Sessile serrated adenoma, AN: Advanced neoplasia.

## Data Availability

Data are available on request due to restrictions, e.g., privacy or ethical. The data presented in this study are available on request from the corresponding author. The data are not publicly available since the results of this study were synthesized using publicly available data from other studies.

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
