# Peer review of "Risk of Metachronous Neoplasia with High-Risk Adenoma and Synchronous Sessile Serrated Adenoma: A Systematic Review and Meta-Analysis"

_diagnostics, 2023, doi:10.3390/diagnostics13091569_

Round 1
Reviewer 1 Report
I have read with great interest the manuscript entitled “Risk of Metachronous Neoplasia with High-Risk Adenoma and Synchronous Sessile Serrated Ade-noma: A Systematic Review and Meta-Analysis”. It adds some interesting information that would help in the development of the future surveillance guidelines. However, although the meta-analysis is extremely interesting, I miss the authors did not evaluate CRC incidence as the main outcome. Metachronous adenoma is just a secondary outcome and an increased risk should not modify the surveillance recommendations. This is the most important limitation and authors should acknowledge it in the discussion.
Abstract:
Although the results are relevant, I do not think the authors can conclude: “Therefore, shorter surveillance intervals should be considered in patients with 37 high-risk adenoma and synchronous sessile serrated adenomas compared to those with high-risk 38 adenoma alone”. As previously commented the main outcome is CRC incidence and authors have not evaluated this outcome.
Introduction.
In paragraph 2 the authors state the risk of progression of adenomas to CRC. Although this was a point of interest in the 70s when adenomas were not resected, this is not the point of interest. The real matter is the risk of CRC detection after adenoma (or serrated lesion) resection. Obviously, it is not strictily related to the risk of progression of an already resected lesion. I recommend the authors to change this paragraph.
In the fourth paragraph, the authors state that “There is currently no specific recommendation for the surveillance of colon polyps 85 based on the presence of different high-risk polyp histology”. In fact, all the guidelines evaluate these characteristics. In some cases, they include them, in other cases, they do not include them because they do not increase the CRC risk.
Methods:
Although they refer to the PRISMA statement, authors do not describe they are performing as systematic review and meta-analysis
Results:
The quality assessment should be included as a table in the results.
The numbers do not fit between table 1 (Characteristics of studies included in the meta-analysis.) the first paragraph and the secondary outcome. On account of the number expressed in table 1, all patients included in the systematic review (2490) had high risk adenomas, 2068 without SSA and 422 with SSA. I do not find where are the patients with low risk adenomas. It must be revised and clarified.
Author Response
Dear reviewer,
Thank you for your time and valuable suggestions. we have made changes to the manuscript as recommended. Kindly let us know if there is anything we could do further. Thank you
I have read with great interest the manuscript entitled “Risk of Metachronous Neoplasia with High-Risk Adenoma and Synchronous Sessile Serrated Adenoma: A Systematic Review and Meta-Analysis”. It adds some interesting information that would help in the development of the future surveillance guidelines. However, although the meta-analysis is extremely interesting, I miss the authors did not evaluate CRC incidence as the main outcome. Metachronous adenoma is just a secondary outcome and an increased risk should not modify the surveillance recommendations. This is the most important limitation and authors should acknowledge it in the discussion.
Dear reviewer,
Thank you for your suggestion. We agree with your observation and have reported as a limitation. Please refer to page 10, line 339-341.
Abstract:
Although the results are relevant, I do not think the authors can conclude: “Therefore, shorter surveillance intervals should be considered in patients with 37 high-risk adenoma and synchronous sessile serrated adenomas compared to those with high-risk 38 adenoma alone”. As previously commented the main outcome is CRC incidence and authors have not evaluated this outcome.
Dear reviewer,
We agree with your suggestion. We have changed the conclusion to shorter surveillance may be considered instead of should be considered. Please refer to page 1 line 37.
Introduction.
In paragraph 2 the authors state the risk of progression of adenomas to CRC. Although this was a point of interest in the 70s when adenomas were not resected, this is not the point of interest. The real matter is the risk of CRC detection after adenoma (or serrated lesion) resection. Obviously, it is not strictly related to the risk of progression of an already resected lesion. I recommend the authors to change this paragraph.
Dear reviewer, we intend to state that the risk of CRC in patients who have had synchronous (high risk and SSA) polyps with two different pathway for cancer. And not necessarily to state risk of CRC from previously resected polyps. To state this better, we have changed our manuscript. Please refer to page 2, line 70-71.
In the fourth paragraph, the authors state that “There is currently no specific recommendation for the surveillance of colon polyps 85 based on the presence of different high-risk polyp histology”. In fact, all the guidelines evaluate these characteristics. In some cases, they include them, in other cases, they do not include them because they do not increase the CRC risk.
Methods:
Although they refer to the PRISMA statement, authors do not describe they are performing as systematic review and meta-analysis
Dear reviewer, please refer to page 3, line 103. We have made changes as suggested. Thank you.
Results:
The quality assessment should be included as a table in the results.
Dear reviewer, please note that the quality assessment is provided as a supplementary file as Supplementary table 1. However, if the journal agrees, we can include it as Table 2 in the main manuscript. Will make the changes once confirmed by the journal. Thank you.
The numbers do not fit between table 1 (Characteristics of studies included in the meta-analysis.) the first paragraph and the secondary outcome. On account of the number expressed in table 1, all patients included in the systematic review (2490) had high risk adenomas, 2068 without SSA and 422 with SSA. I do not find where are the patients with low-risk adenomas. It must be revised and clarified.
Dear reviewer,
Thank you for your suggestions. We have clarified the numbers. There were 5628 patients who initially underwent screening colonoscopies. However, only 2490 patients who had either high risk adenoma or both HRA and SSA. The second set of patients were included in the calculation of primary outcome. This is the reason why you see higher number of patients in the low-risk adenoma pooled estimates. We hope that clarifies the numbers. Please refer to page 6 lines 201-204. Thank you.
Reviewer 2 Report
Dear Author,
I was delighted to review this systematic review and meta-analysis. The introduction, methods and results are very well written. There appears to have been a rigorous search criterion and the review attempts to answer an interesting question. My only major concern is the discussion which has not been properly proofed with multiple typos, grammatical errors, repetitive language and insufficient referencing. I believe this paper could be improved by significantly revising the discussion.
General points to address:
One limitation of your study is that you haven’t been able to look for significant confounding. For example, do patients with synchronous high-risk adenoma and sessile serrated polyps simply have a higher index polyp burden (number of polyps). It’s difficult to conclude that the combination of synchronous adenoma/ sessile serrated polyps is truly acting in a synergistic fashion to increase metachronous neoplasia risk. Is there any published evidence which gives a biological basis for this synergy?
No mention of dysplasia in sessile serrated polyps. In our practice we would qualify sessile serrated polyps ≥10mm or containing any grade of dysplasia as high-risk lesions.
Inconsistent use of colorectal cancer or CRC. Use one or the other.
Point-by-point suggested changes:
Results
Table 1 legend - is it possible to have the median time between index and surveillance colonoscopy as this would be more useful than mean.
Table 1 - it would be useful to display percentages, perhaps in brackets, to make it easier to immediately ascertain the percentage of males in each study, the percentage of those with index high risk adenoma and SSP etc. This is particularly the case for the final two columns which I believe are the primary outcome variables. Could the final column have “on surveillance colonoscopy” as per the penultimate column added into the heading for clarity?Patient characteristics - again no percentages included.Secondary outcomes - could the word “and” be replaced with “versus” or “as compared to” in the final sentence describing figure 4.
Discussion
There is a lack of referencing. For example, early on you say the “incidence of CRC decreased by 1% per year from 2011 to 2019”. You should reference where this information came from. “Previous studies have suggested higher risk of CRC in those with HRA” - what previous studies? “First degree relatives with HRA leads to higher risk of CRC” - must cite a source for this claim. Guidelines are mentioned several times, but no reference given to these guidelines.Line 244 - replace colon cancer with colorectal cancer.
Line 246 does not make sense and there is a typo of “maily” for mainly - please revise.
Line 253 - should be risk “factors”.
Line 256 “polyp” rather than polyps.
Line 257 - replace “based on” with “by”.
Line 290 - no need to give US multi society task force and abbreviation as previously given above.
Line 291 - replace “in 7-10 years” with “7-10 years from index colonoscopy”.
Line 298 needs revised - most recent to what? Should possibly read “another similar meta-analysis was recently conducted” or similar.
Line 299 - reference 30 should be given after the description of this paper rather than the sentence below comparing to your own paper.
Line 308 - should dysplasia be replaced with high grade dysplasia? Likewise in line 309.
Line 314 typo “rsults”.
Line 316 – replace “are due to” with “can be attributed to”.
Line 318 to 320 doesn’t read correctly – words missing and grammatical errors. Revise.
Line 315 to 324 – repetitive language, same point made multiple times, be more concise.
Line 325 – sentence doesn’t make sense.
Line 332 – I think your study has shown an increased risk of metachronous neoplasia, not colorectal cancer.
Line 336 – replace “and their increased risk” with “and the possibility of increased risk of progressing to colorectal cancer”.
Line 341 – type “ednoscopists”.
Line 346 – you have not shown that these individuals are at higher risk of CRC. You need to say something like “these individuals may be at increased risk of CRC”.
Line 353 – typo “no” instead of “of”.
Kind regards
Author Response
Dear reviewer,
Thank you for all your valuable suggestions. we have made changes to the manuscript as suggested. Kindly let us know if there is anything else we could do to make the manuscript better. Thank you.
Best wishes
All authors.
I was delighted to review this systematic review and meta-analysis. The introduction, methods and results are very well written. There appears to have been a rigorous search criterion and the review attempts to answer an interesting question. My only major concern is the discussion which has not been properly proofed with multiple typos, grammatical errors, repetitive language and insufficient referencing. I believe this paper could be improved by significantly revising the discussion.
Dear reviewer,
Thank you for your suggestions. We have made significant improvement to the manuscript to correct the grammar. We would be happy to make further changes as needed. Please review the changes and kindly let us know.
General points to address:
One limitation of your study is that you haven’t been able to look for significant confounding. For example, do patients with synchronous high-risk adenoma and sessile serrated polyps simply have a higher index polyp burden (number of polyps). It’s difficult to conclude that the combination of synchronous adenoma/ sessile serrated polyps is truly acting in a synergistic fashion to increase metachronous neoplasia risk. Is there any published evidence which gives a biological basis for this synergy?
Dear reviewer,
Thank you very much for your valuable suggestion. We agreed with your observation. We have updated our limitations in the manuscript. Please refer to page. the total number of polyp burden in the included population can play a role in the development of subsequent colon cancer. However, this data was not reported in the included studies. Please refer to page 10, line 352-354.
No mention of dysplasia in sessile serrated polyps. In our practice we would qualify sessile serrated polyps ≥10mm or containing any grade of dysplasia as high-risk lesions.
Dear reviewer,
We agree with your observation. Unfortunately, we could not perform pooled estimates for presence or absence of dysplasia in patients with SSAs due to lack of data.
Inconsistent use of colorectal cancer or CRC. Use one or the other.
Dear reviewer, we have made changes to the manuscript as suggested throughout the manuscript, thank you.
Point-by-point suggested changes:
Results
Table 1 legend - is it possible to have the median time between index and surveillance colonoscopy as this would be more useful than mean.
Dear reviewer, due to lack of consistent reporting, we decided to use mean duration between screening and surveillance colonoscopies. For now, we prefer to keep mean. However, if we will make further to see if we can delineate the median from the included studies.
Table 1 –
it would be useful to display percentages, perhaps in brackets, to make it easier to immediately ascertain the percentage of males in each study: the percentage of those with index high risk adenoma and SSP etc. This is particularly the case for the final two columns which I believe are the primary outcome variables. Could the final column have “on surveillance colonoscopy” as per the penultimate column added into the heading for clarity?
Dear reviewer, thank you for this very valuable suggestion. We have made changes as recommended. Please refer to page 6, Table 1.
Patient characteristics - again no percentages included.
Thank you Please refer to page 6, line205-206.
Secondary outcomes - could the word “and” be replaced with “versus” or “as compared to” in the final sentence describing figure 4.
Dear reviewer, changes were made as suggested. Please refer to page 8, line 260.
Discussion
There is a lack of referencing. For example, early on you say the “incidence of CRC decreased by 1% per year from 2011 to 2019”. You should reference where this information came from. “Previous studies have suggested higher risk of CRC in those with HRA” - what previous studies? “First degree relatives with HRA leads to higher risk of CRC” - must cite a source for this claim. Guidelines are mentioned several times, but no reference given to these guidelines.
Line 244 - replace colon cancer with colorectal cancer.
Dear reviewer, we have made the change. Please refer to page 8, line 269.
Line 246 does not make sense and there is a typo of “maily” for mainly - please revise.
Dear reviewer, we have made the change. Please refer to page 8, line 271.
Line 253 - should be risk “factors”.
Dear reviewer, we have made the change. Please refer to page 8, line 278.
Line 256 “polyp” rather than polyps.
Dear reviewer, we have made the change. Please refer to page 8, line 281.
Line 257 - replace “based on” with “by”.
Dear reviewer, we have made the change. Please refer to page 8, line 281.
Line 290 - no need to give US multi society task force and abbreviation as previously given above.
Dear reviewer, we have made the change. Please refer to page 8, line 330.
Line 291 - replace “in 7-10 years” with “7-10 years from index colonoscopy”.
Dear reviewer, we have made the change. Please refer to page 8, line 331.
Line 298 needs revised - most recent to what? Should possibly read “another similar meta-analysis was recently conducted” or similar.
Dear reviewer, we have made the change. Please refer to page 8, line 330.
Line 299 - reference 30 should be given after the description of this paper rather than the sentence below comparing to your own paper.
Dear reviewer, we have made the change. Please refer to page 8, line 339-341.
Line 308 - should dysplasia be replaced with high grade dysplasia? Likewise in line 309.
Dear reviewer, we have made the change. Please refer to page 8, line 349-351.
Line 314 typo “rsults”.
Dear reviewer, we have made the change. Please refer to page 8, line 366.
Line 316 – replace “are due to” with “can be attributed to”.
Dear reviewer, we have made the change. Please refer to page 8, line 373.
Line 318 to 320 doesn’t read correctly – words missing and grammatical errors. Revise.
Dear reviewer, we have made the change. Please refer to page 8, line 370-372.
Line 315 to 324 – repetitive language, same point made multiple times, be more concise.
Dear reviewer, we have made the change. Repetitive point has been removed.
Line 325 – sentence doesn’t make sense.
Dear reviewer, we have made the change. The statement has been removed.
Line 332 – I think your study has shown an increased risk of metachronous neoplasia, not colorectal cancer.
Dear reviewer, we have made the change. We agree and made changes to the statement to reflect meta-chronous neoplasia. Please see page 10, line 385-387.
Line 336 – replace “and their increased risk” with “and the possibility of increased risk of progressing to colorectal cancer”.
Dear reviewer, we have made the change. We agree and made changes to the statement. Please see page 10, line 390.
Line 341 – type “ednoscopists”.
Dear reviewer, we have made the change. We agree and made changes to the statement. Please see page 10, line 395.
Line 346 – you have not shown that these individuals are at higher risk of CRC. You need to say something like “these individuals may be at increased risk of CRC”.
Dear reviewer, we have made the change. We agree and made changes to the statement. Please see page 10, line 399.
Line 353 – typo “no” instead of “of”.
Dear reviewer, we have made the change. We agree and made changes to the statement. Please see page 10, line 406.
Kind regards
Round 2
Reviewer 1 Report
No comment